# Scutellaria Barbata inhibits epithelial-mesenchymal transformation through PI3K/AKT and MDM2 thus inhibiting the proliferation, migration and promoting apoptosis of Cervical Cancer cells

Guobing Wang[1�], Congchao Jia[2�], Dexin Wang[2,3,4], Lei Liang[2,3,4], Lan Li[2,3,4], Gang Tian[2,3,4]*, Wenqi Feng [1]*

**1** Medical Research Laboratory, YiBin Hospital of T.C.M, Yibin, China, **2** Department of Laboratory Medicine, The Affiliated Hospital of Southwest Medical University, Luzhou, China, **3** Sichuan Province Engineering Technology Research Center of Molecular Diagnosis of Clinical Diseases, Luzhou, China, **4** Molecular Diagnosis of Clinical Diseases Key Laboratory of Luzhou, Luzhou, China

९ These authors contributed equally to this work and share first authorship.
* tiangang@swmu.edu.cn (GT); moonly1981@163.com (WF)

## Abstract

### Background

Cervical cancer is the fourth most common malignant tumor among women with high morbidity and mortality. We found the active ingredients, action targets and pathways of Scutellaria Barbata against cervical cancer cells, and verified its effects and mechanisms on the proliferation, migration and apoptosis of cervical cancer cells through cellular experiments.

### Materials and methods

The potential effective components of Scutellaria Barbata were obtained by data mining. CCK 8 experiments were used to verify the optimal action time and concentration of the key active components, Baicalin and Wogonin on Hela cells. Transwell Experiments to verify the migration rate of Hela cells at different time points of action. Hela cell apoptosis was determined by a Tunel assay. RT-qPCR and Western Blot detection of related genes and protein expression after the effects of different concentrations of Baicalein and Wogonin on Hela cells.

### Results

Obtained 27 core intersecting targets. Four genes associated with survival of cervical cancer patients were selected by survival by univariate analysis: EGFR, TNF-α, VEGFA and MDM2 (P<0.1). GO and KEGG pathway enrichment analysis showed the four genes played positive regulatory roles, mainly enriched in cellular extracellular zone and cytoplasm. Pathways include cancer pathways, proteoglycans and PI3K/AKT signal. CCK 8 experiments showed that the proliferation rate of Hela cells was significantly lower than

**Data availability statement:** The data is available in the Figshare repository at the following DOI: (10.6084/m9.figshare.28254122)

**Funding:** This study was supported by grants from the Luzhou Science and Technology Department Applied Basic Research Program (No: 2022-JYJ-145), the Sichuan Province Science and Technology Department of foreign (border) high-end talent introduction project (No: 2023JDGD0037), Sichuan Provincial Medical Association (No: Q22027), YiBin Science and Technology Department Social Development Projects (No:2022SF004), and Medical Scientific Research Project of Yibin Municipal Health Commission (No:2023YW026).

**Competing interests:** The authors declare that the research was conducted in the absence of any commercial or financial relationships that could be construed as a potential conflict of interest.

that of the control group, and the optimal drug concentration was 10 mg/mL (P <0.001). The migration assay showed that the incubation of the medium with a mixture of 10 mg/mL each or 72h of 5 ml/mL each significantly inhibited the migration of Hela cells (P <0.001). Tunel results showed that single-agent Baicalein or Wogonin with 10 mg/mL had the highest rate of apoptosis on Hela cells after culturing the cells for 72 h. Baicalein or Wogonin (10 mg/mL + 10 mg/mL) had the highest apoptosis ratio in Hela cells after 72h and a higher combination ratio than in the single agent group (P <0.001). The difference between the single drug group and the corresponding low dose group was significant (P<0.001). After Hela cells were cultured for 72 h, the mRNA and protein expressions corresponding to EGFR, TNF-α, VEGFA, PI3K, AKT and MDM2 were decreased in the experimental group compared with the control group as confirmed by RT-PCR and Western blot, and the extent of the decrease was more obvious with the increase of the drug concentration, and under the condition of the same drug dosage, the effect of the Baicalein group was more pronounced than that of the Wogonin group, and the effect was more obvious in the two-drug group compared with the single-drug group (P<0.05).

## Conclusion

After combining with EGFR, TNF-α and VEGFA, the active ingredient of Sculellaria Barbata can regulate the PI3K/AKT signaling pathway and down-regulate the expression of MDM2 gene to inhibit the epithelial-mesenchymal transition, which can inhibit cervical cancer cell proliferation, migration, and promote their apoptosis. Sculellaria Barbata is a potential therapeutic candidate for cervical cancer.

## 1. Introduction

As a common malignant tumor among women, cervical cancer poses a serious risk to women's physical and mental wellbeing because of its high malignancy, incidence, and recurrence rate after surgery [1]. In 2018, there were about 570,000 new cases of i worldwide, accounting for 3.15% of the incidence of all malignant tumors, and about 310,000 deaths, accounting for 3.26% of the total number of deaths from all malignant tumors. The incidence rate in China is about 3 ‰, and the incidence rate accounts for 18.6% of the global malignant tumors. It has been shown that persistent high-risk HPV infection is closely associated with cervical cancer and precancerous lesions, which effectively reduces the clinical underdiagnosis and misdiagnosis rate and improves the cervical cancer accuracy [2]. However, combined screening also has certain false-negative and false-positive rates, and the peak age of cervical cancer patients is showing a younger trend [3]. Therefore, preventing and treating cervical cancer remains a clinical challenge.

Currently, surgery, radiotherapy, and chemotherapy are the main treatments for cervical cancer, and Chinese medicine can also be used as adjuvant treatment [4]. The advantages of surgical treatment are that the cancer can be completely removed in one operation in early stage cases, the treatment time is short, and the ovarian function of normal patients can be preserved [5]. However, some patients may have serious complications during surgical treatment, slow postoperative recovery, and even sequelae [6]. Radiotherapy is applicable to all patients with cervical cancer, and while killing cancer cells, radiotherapy can also damage normal tissues in the irradiated field, with complications in the rectum and bladder being the most common [7]. Chemotherapy is a treatment method targeting the systemic level

and is mainly used for advanced cases of cervical cancer or combined surgical treatment and radiation therapy, and can also be used to treat recurrent tumors [8]. However, chemotherapy has certain toxic side effects, including nausea, vomiting, loss of appetite and gastrointestinal and bone marrow suppressive reactions such as decreased white blood cells and platelets [9]. In contrast, TCM with surgery, radiation and chemotherapy for cervical cancer has obvious advantages, not only in reducing the side effects caused by radiation therapy or chemotherapy, but also in improving the efficacy and accelerating the recovery of patients, so it is worth further exploring its mechanism of action [10].

There have been a growing number of studies showing the anti-tumor properties of Scutellaria Barbata, especially in cervical cancer treatment [11]. Antitumor and antiviral effects of Scutellaria Barbata are gaining attention in modern medicine and are of great significance in revealing the mechanism of anti-tumor effects and screening new anti-tumor drugs [12]. Studies have shown that the main medicinal active components of Scutellaria Barbata are Baicalin and Wogonin [13]. Baicalin and Wogonin were isolated and purified from Scutellaria Barbata by high performance liquid chromatography (HPLC). These two effective components did not contain endotoxin. It is difficult to elucidate the metabolic pathways and molecular mechanisms of TCM in vivo because of its many components and complex metabolic processes [14]. Therefore, finding a research model that follows the TCM theory of starting from the whole and treating with evidence, which is in line with the requirements of TCM theory and prescription, and establishing a research model with multiple component, target and pathway action characteristics is the key to the modernization of TCM research [15]. With the rapid development of systems biology and computer technology network pharmacology came into being, network pharmacology is a new way of thinking about drug design based on network biology and multidirectional pharmacology, which is consistent with the view that TCM diagnoses and treats diseases from a holistic perspective and TCM regulates diseases from a holistic level [16]. Therefore, we mainly focused on network pharmacology analyses of Scutellaria Barbata as a potential treatment for cervical cancer.

## 2. Materials and methods

### 2.1 Prediction the mechanism of action of Scutellaria Barbata on cervical cancer by network pharmacology

**2.1.1 The chemical composition of Scutellaria Barbata collection and screening.** The potential active components of Scutellaria Barbata were screened by TCMSP (https://old. tcmsp-e.com/tcmsp.php), a systematic pharmacology platform for Chinese herbal medicines, and STITCH (https://stitch.embl.de/) database for chemical-protein interactions, and the chemical formulas were constructed. Oral bioavailability (OB) ≥30% and drug-like properties (DL) ≥0.18 in the formulation of Hemerocallis were used to screen the active compounds. Component target prediction of hemicrania was performed by Swiss Target Prediction (http://www.swisstargetprediction.ch) and Drug Target Database (DrugBank, https://www.drugbank.com/) (Fig 1).

**2.1.2 Acquisition of targets of action in cervical cancer.** The human Gene-disease-related database, OTP (https://www.opentargetsplatform.org/), and GeneCards (https://www.genecards.org/) offer comprehensive information on all known and predicted human genes in various aspects such as the genome, proteome, transcription, genetics, and function. In this study, we conducted a search using the GeneCards database and the OTP database to identify targets associated with cervical cancer disease. The results from both databases were merged, structured, and refined based on a predefined threshold to pinpoint the targets relevant to cervical cancer.

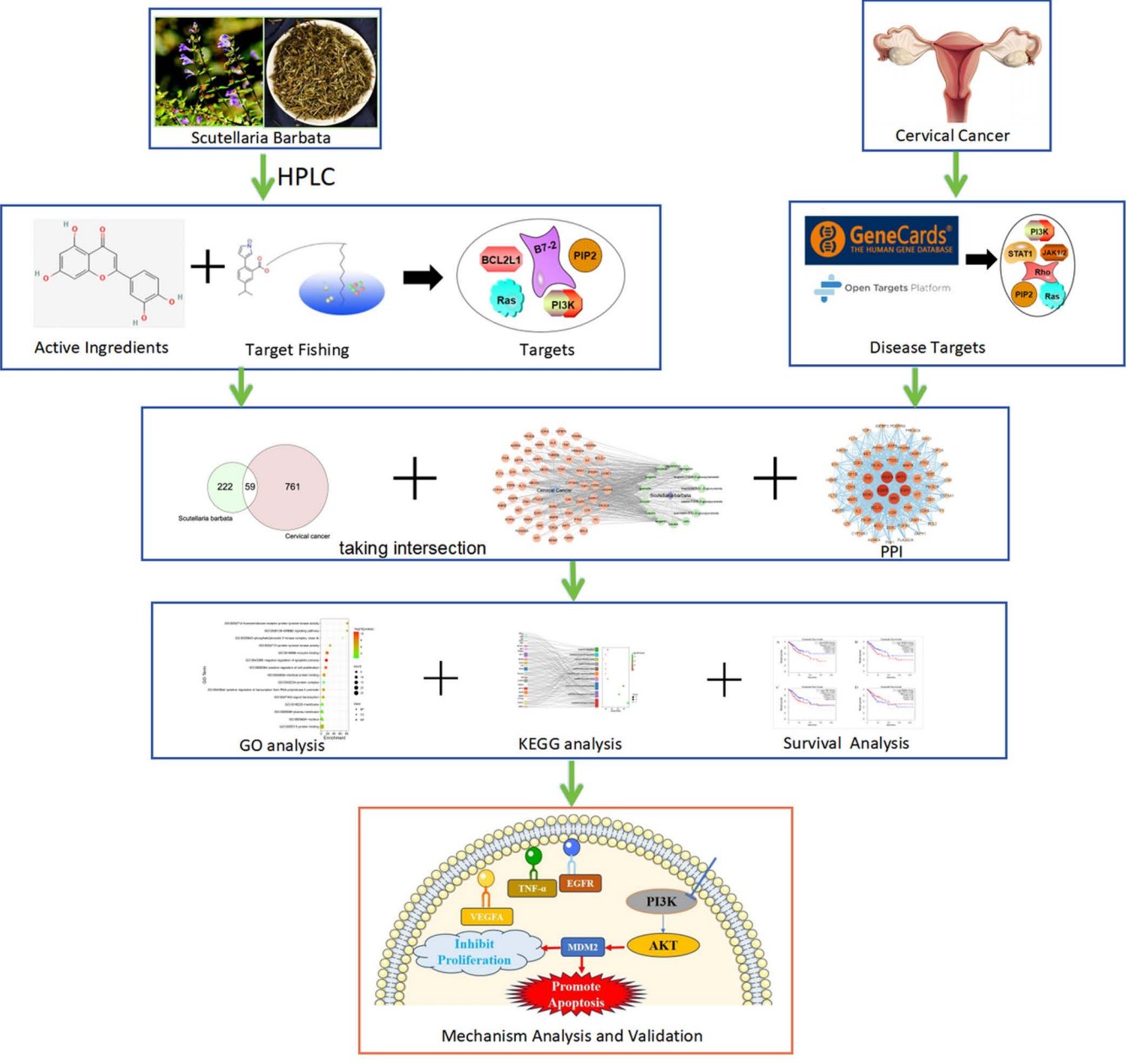

**Fig 1. Flow chart of collection and screening of chemical components of Scutellaria Barbata.**

**2.1.3 Construction of a drug-active ingredient-target gene-disease network.** The target genes corresponding to the active ingredients of Scutellaria Barbata and the target genes related to cervical cancer were introduced into venny2.1.0 (https://bioinfogp.cnb.csic.es/tools/venny/) to obtain the intersection between the two, and the common genes of the two were the key targets of Scutellaria Barbata against cervical cancer. The Cytoscape3.7.2 software was used to construct the "drug-active ingredient-target gene-disease" relationship network to explore the mechanism of Scutellaria Barbata action against cervical cancer.

**2.1.4 Construction of the relevant action network of key target proteins and the screening of core targets.** The common genes obtained above were analyzed using the String platform (https://string-db.org/Version 10.5) database, and then the protein-protein interaction network (Protein-protein interaction, PPI) was constructed. PPI searches between known proteins and predicted proteins. The common target of Scutellaria Barbata and cervical cancer was entered into STRING11.0 (https://stringdb.org/) database, the data screening condition was limited to "multiple proteins", the species was set to "human", the placement level was 0.95, free nodes were excluded, and the other parameters were default values. The PPI network of target site was built, the results were saved as TSV file, and the saved results were imported into Cytoscape.3.7. 2 Software to obtain the two highly interacting mechanism clusters. Topological analysis of the network was performed using the Network Analyzer plugin to screen out the top 10 core targets for cervical cancer treatment based on the degree (degree ≥31) values. Univariate survival analysis of the selected target genes was performed one by one, and the core gene targets highly correlated with the survival rate of cervical cancer patients were selected at P <0.1.

**2.1.5 GO functional enrichment analysis and KEGG pathway analysis.** Gene ontology (gene ontology, GO) functional enrichment analysis and Kyoto Encyclopedia (Kyoto Encyclopedia of Genes and Genomes, KEGG) pathway analysis can identify key genes related to the prognosis of cervical cancer for further experimental studies. The 59 intersection genes were imported into the DAVID 6.8 database for GO functional enrichment analysis and KEGG pathway analysis. The role of the target of cervical cancer in GO function enrichment was analyzed from three modules: biological processes, molecular function and cellular components. Entries with P <0.05 were selected for bars using GraphPad Prism 8.3.0 and KEGG pathway bubble plots using R software. The "core compound-target-pathway" network was constructed using Cytoscape 3.7.2 software. In the network graph, each node represents different compounds, targets or pathways, and the edges represent the interaction relationships between compounds, targets or pathways.

## 2.2 Experimental cell-level validation and action pathway validation of anti-cervical cancer mechanism

**2.2.1 Cell Culture.** Human cervical cancer cells (Hela) were presented by Department of Oncology, Affiliated Hospital of Southwest Medical University. Cells were cultured at 37°C, 5% CO2, and 95% air in a medium comprising 90% DMEM medium (Thermo Fisher, USA), 10% fetal bovine serum (Thermo Fisher, USA), and 1% penicillin-streptomycin. Control group: Hela cells + complete culture medium and experimental group corresponding time; Experimental group: Hela cells were treated with +0, 0.25, 0.5, 1, 2, 5, 10 mg/mL Baicalin (Sigma-Aldrich, USA) or Wogonin (Sigma-Aldrich, USA)for 0, 24, 48, 72 hours.

**2.2.2 Assay for cell viability.** A 96-well plate was seeded overnight with HeLa cells. HeLa cells were treated with concentration gradient of Baicalin (0, 0.25, 0.5, 1, 2, 5, 10 mg/mL), or the concentration gradient of Wogonin (0, 0.25, 0.5, 1, 2, 5, 10 mg/mL), the intervention time set for 0, 24, 48, 72 hours. In the following step, cell viability was measured using a Microplate Reader by using Cell Counting Kit8 (CCK-8).

**2.2.3 Transwell migration.** According to the manufacturer's protocol, 24-well, 8.0 micron pore membranes (Corning, USA) were used. In the upper chamber, $1 \times 10^5$ cells were seeded in 200 μL of serum-free medium, and a chemoattractant of 800 μL of complete medium was added simultaneously to the lower chamber. Using cotton swabs, we removed the cells remaining on top of the membrane after incubation at 37°C for 24 hours, and the cells on the lower surface were those that had migrated. The cells that passed through the filter

were photographed by inverted fluorescence microscope (Olympus, Japan) after being fixed with 4% paraformaldehyde and stained with 0.1% crystal violet solution.

**2.2.4  TUNEL assay.**  In addition to the TUNEL assay, we measured apoptosis in cultured Hela cells fixed by 4% paraformaldehyde on coverslips. Under fluorescence microscopy, apoptotic Hela cells were observed using an in situ cell death detection kit (Corning Incorporated, China).

**2.2.5  qRT-PCR.**  Invitrogen TRIzol reagent was used to extract total RNA from cultured cells, and 1 μg of total RNA was synthesized into cDNA.Target gene expression levels were assessed using SYBR R Premix Ex TaqTM II (Takara, Japan) followed by GAPDH normalization. Detailed primer sequences can be found in Supplementary S1 Table.

**2.2.6  Western Blot.**  10% SDS PAGE was used to separate the total protein samples and PVDF membranes were used as the transfer medium. 5% nonfat milk was used to block membranes, followed by primary antibodies and secondary antibodies that were HRP-tagged. Samples were quantified using Western blotting detection kit ECL Prime (GE Healthcare, USA) after culturing in TBST solution.

**2.2.7  Statistical analysis and mapping.**  We analyzed the data using SPSS 25.0 and provided the results in the following form: mean ± standard deviation($\bar{X}$ ±s); The difference between the two groups was compared using a T-test, as One-way ANOVA indicated that there was a statistical difference between groups at $P<0.05$.

## 3.  Results

### 3.1  Results of the network pharmacology analysis

**3.1.1  Chemical composition and target information.**  A total of 14 active components were obtained and chemical formulas were constructed (Fig 2 and Table 1). 222 predicted targets were obtained by Swiss Target Prediction and the drug target database DrugBank.

**3.1.2 Related targets of cervical cance.**  761 cervical cancer-related targets were obtained from the GeneCards and OTP databases. Using 222 predicted targets and 761 related targets of cervical cancer, the venney 2.1.0 online analysis tool was used to draw a visual Venn diagram of venney 2.1.0,59 intersection target genes were obtained.(Fig 3).

**3.1.3 Construction of " Scutellaria Barbata - active ingredient - target gene - cervical cancer " network relationship.**  Target genes corresponding to Scutellaria Barbata's active ingredient and the target genes related to cervical cancer were compared for drug-disease associations, and the common genes between them were obtained as the key targets of Scutellaria Barbata against cervical cancer. Cytoscape 3.7.2 was used to construct the "Scutellaria Barbata-active ingredient-targets gene-cervical cancer" network to investigate the mechanism of action of Scutellaria Barbata against cervical cancer(Fig 4).

**3.1.4  Protein interaction network and access to core targets.**  String database is a protein interaction network database based on public database and literature information, which can provide comprehensive information of protein interaction. The common target genes of Scutellaria Barbata and uterine neck cancer were entered into String database for PPI analysis, and then the obtained data were imported into Cytospace, and 27 core targets were obtained according to degree ≥ 24.(Fig 5 and Table 2). Univariate survival analysis analyzed the 27 core targets, and the core gene targets were highly associated with the survival of cervical cancer patients: EGFR, MDM2, TNF and VEGFA (Fig 6).

**3.1.5  GO functional enrichment analysis and KEGG pathway enrichment analysis.**  DAVID6.8 was used to analyze GO functional enrichment of 59 intersecting genes, and the top 10 CC ranked entries were plotted in a bar graph after sorting the number of gene enrichment Count with $P<0.05$. Cellular responses to chemical stimuli, organic substances, and

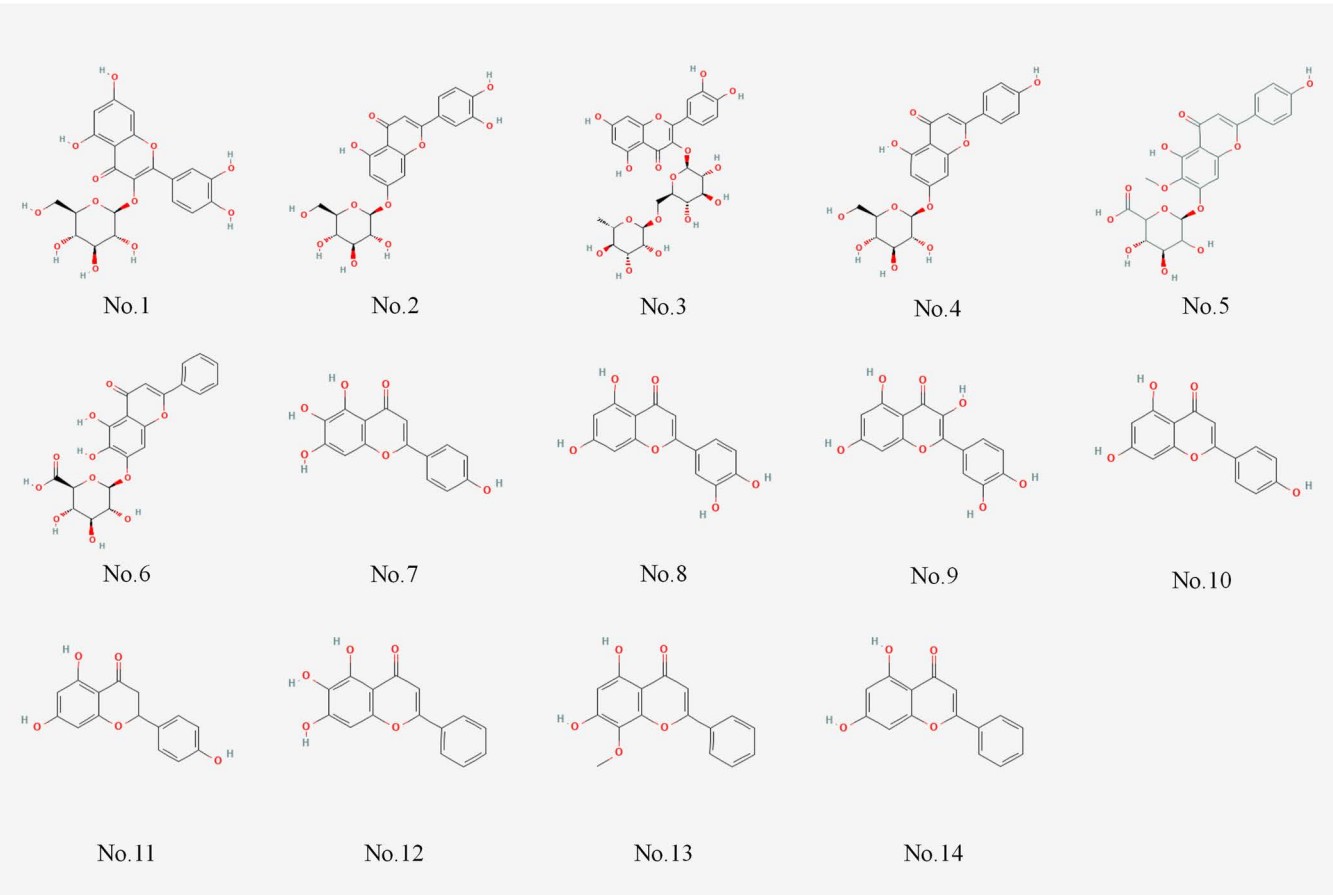

**Fig 2. Chemical composition and target information of Chinese medicine.**

**Table 1. Composition of Scutellaria Barbata.**

| | Compound | Formulas | Molecular Weight |
|---|---|---|---|
| 1 | quercretin-3-O-β-D-glucopyranside | $C_{21}H_{20}O_{12}$ | 464.10 |
| 2 | luteilin-7-O-B-D-glucopyranside | $C_{21}H_{20}O_{11}$ | 448.10 |
| 3 | rutin | $C_{27}H_{30}O_{16}$ | 610.15 |
| 4 | apigenin-7-O-B-D-glucopyranoside | $C_{21}H_{20}O_{10}$ | 432.11 |
| 5 | hispidulin-7-O- β-D-glucuronide | $C_{22}H_{20}O_{12}$ | 476.10 |
| 6 | baicalin | $C_{21}H_{18}O_{11}$ | 446.08 |
| 7 | scutellarein | $C_{15}H_{10}O_6$ | 286.05 |
| 8 | luteolin | $C_{15}H_{10}O_6$ | 286.05 |
| 9 | quercetin | $C_{15}H_{10}O_7$ | 302.04 |
| 10 | apigenin | $C_{15}H_{10}O_5$ | 270.05 |
| 11 | naringenin | $C_{15}H_{12}O_5$ | 272.07 |
| 12 | baicalein | $C_{15}H_{12}O_5$ | 272.07 |
| 13 | wogonin | $C_{16}H_{12}O_5$ | 284.07 |
| 14 | chrysin | $C_{15}H_{10}O_4$ | 254.06 |

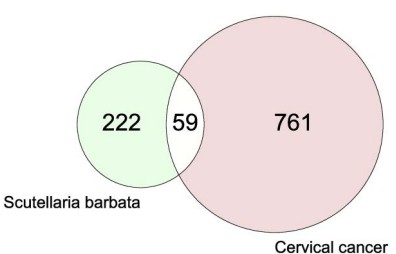

**Fig 3. Scutellaria Barbata active compound target-disease related target visualization Venn diagram.**

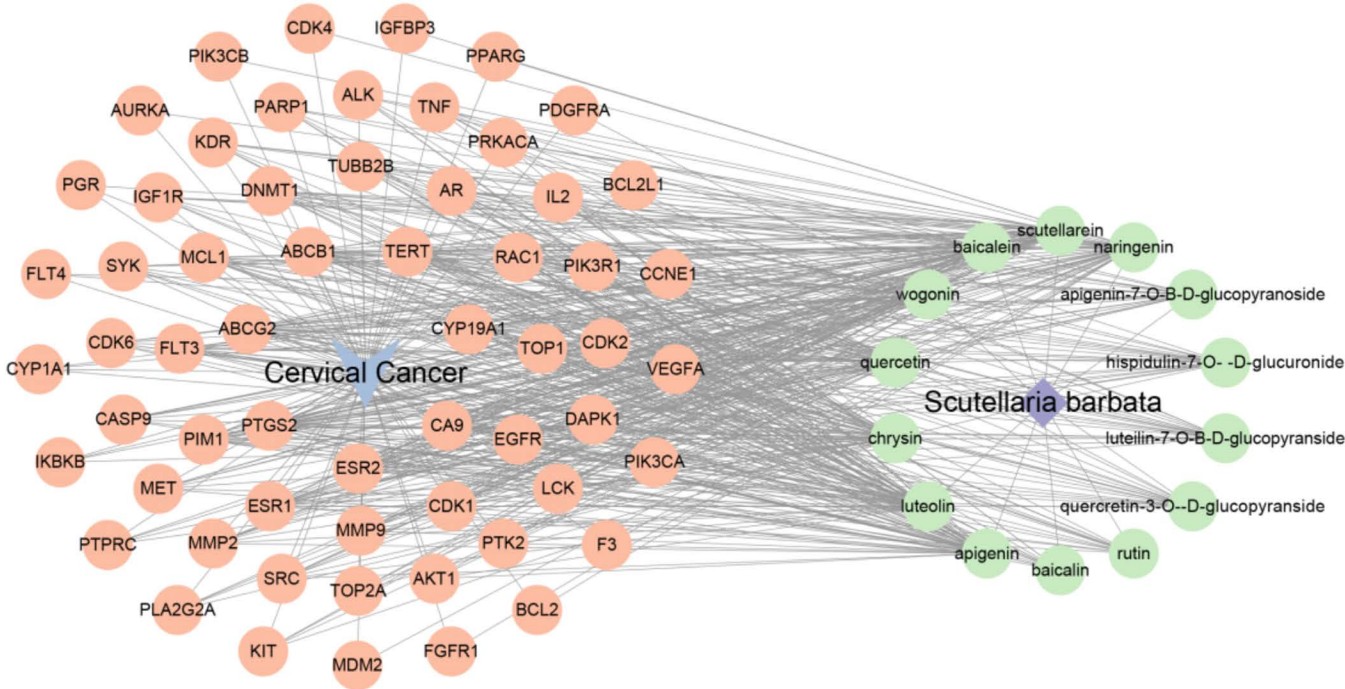

**Fig 4. Drug-active ingredient-target gene-disease network.**

positive regulation of cellular metabolic processes were the main biological processes enriched. From the aspect of cellular components, the main enrichment is in extracellular region, extracellular region part, cytosol, etc. In terms of molecular function, the main enrichment is in organic cyclic compound binding, enzyme binding, receptor binding, protein, etc (Fig 7). DAVID6.8 was used to analyze KEGG pathway in a bar graph after sorting the number of gene enrichment, and 79 signaling pathways were obtained. After screening with P<0.05, R software was used to perform KEGG pathway enrichment analysis on 11 signaling pathways, including pathways involved in cancer, TNF signaling pathways, and the other pathways (Fig 8).

## 3.2 The results of cell level experiments and pathway verification of the anti-cervical cancer mechanism of Scutellaria Barbata

**3.2.1 The anti-proliferative and anti-migratory activity of Baicalin and Wogonin was observed in cervical cancer cells.** The study confirmed that the active ingredients baicalin and wogonin from Scutellaria Barbata can effectively inhibit the proliferation of Hela cells.

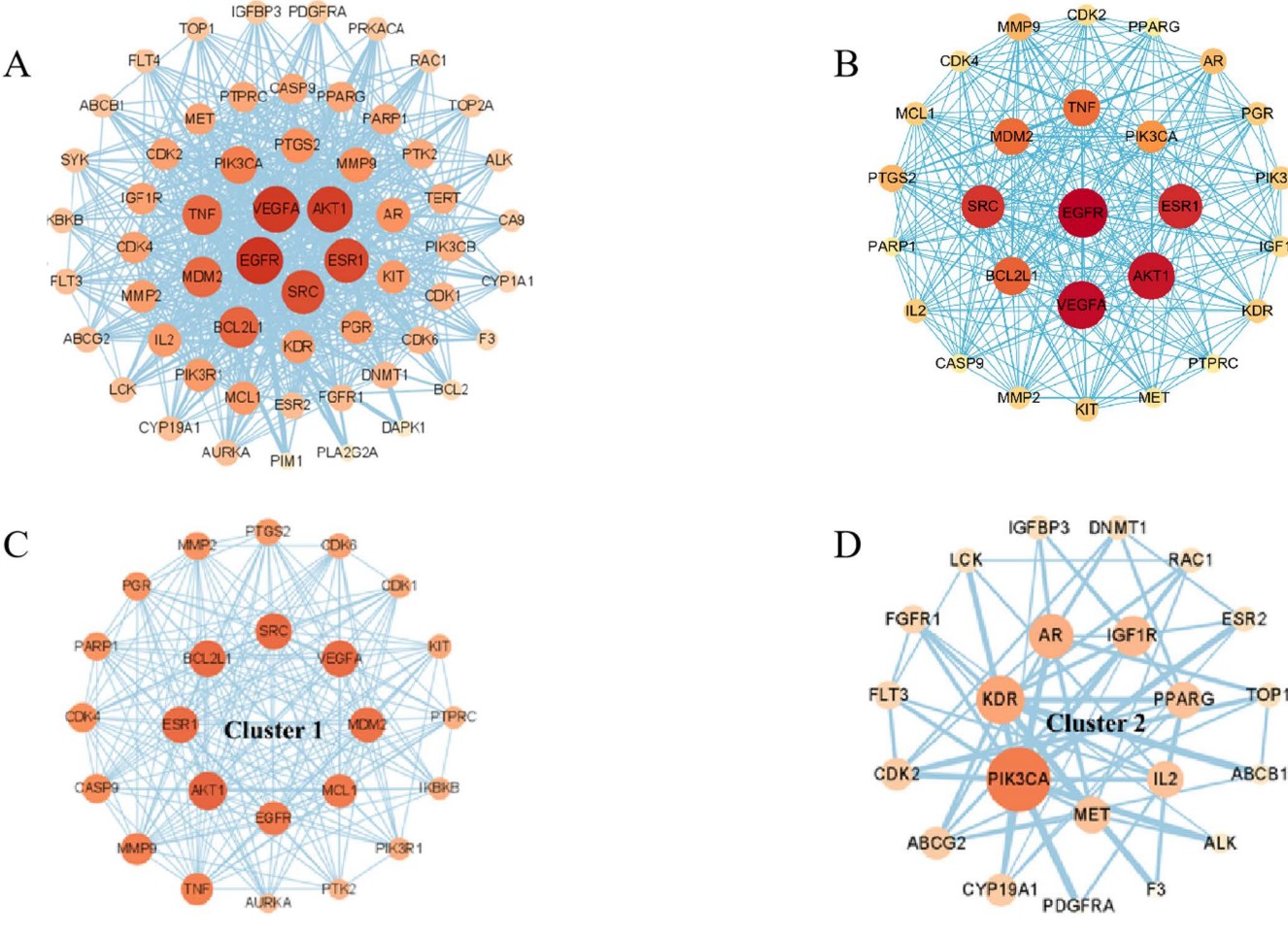

**Fig 5. Protein Interaction Network and Access to Core Targets.**

CCK8 assay results showed a significant decrease in cell proliferation in the experimental group compared to the control group at 48 h and 72 h (P<0.001). This effect was most pronounced at a drug concentration of 10 mg/mL (P<0.001). Over a 72-hour period, there was a significant increase in the inhibition of Hela cell proliferation with both the passage of time and escalating drug concentrations (P<0.01). The suppression of Hela cell proliferation was more pronounced when the drug concentrations reached 5 mg/mL and 10 mg/mL (Fig 9A and B). In the Transwell migration assay, the experimental group exhibited significantly reduced cell migration compared to the control group. Furthermore, simultaneous stimulation with baicalin and wogonin at the same time point resulted in a more pronounced inhibition of cell migration compared to single-drug treatment (P<0.001, Fig 9C).

**3.2.2 Baicalin and Wogonin promote Hela cells apoptosis.** The results of Tunel staining method showed the highest apoptosis ratio to Hela cells after 72 h of single agent Baicalin or Wogonin with 10 mg/mL. Baicalin and Wogonin were respectively cultured in (10 mg/mL + 10 mg/mL) for 12 h, and the apoptosis ratio in the double drug group was higher than that in the single drug group. As the drug concentration increased, the apoptosis ratio increased more significantly in the high dose group than in the corresponding low dose group, and the difference between the single and double drug groups and the corresponding low dose group was statistically significant (P <0.001) (Table 3, Fig 10).

**Table 2. Information of core targets.**

| Number | Gene Symbol | Uniprot ID | Protein Name | OS-p(HR) | Degree |
|---|---|---|---|---|---|
| 1 | EGFR | P00533 | Epidermal growth factor receptor erbB1 | 0.056 | 49 |
| 2 | VEGFA | P15692 | Vascular endothelial growth factor A | 0.032 | 48 |
| 3 | AKT1 | P31749 | Serine/threonine-protein kinase AKT | 0.63 | 47 |
| 4 | ESR1 | P03372 | Estrogen receptor alpha | 0.76 | 45 |
| 5 | SRC | P12931 | Tyrosine-protein kinase SRC | 0.93 | 44 |
| 6 | BCL2L1 | Q07817 | Apoptosis regulator Bcl-X | 0.94 | 40 |
| 7 | TNF | P01375 | TNF-alpha | 0.067 | 39 |
| 8 | MDM2 | Q00987 | E3 ubiquitin-protein ligase Mdm2 | 0.09 | 39 |
| 9 | PIK3CA | P42336 | PI3-kinase p110-alpha subunit | 0.96 | 35 |
| 10 | PTGS2 | P35354 | Cyclooxygenase-2 | 0.16 | 31 |
| 11 | MMP9 | P14780 | Matrix metalloproteinase 9 | 0.41 | 31 |
| 12 | AR | P10275 | Androgen Receptor | 0.44 | 30 |
| 13 | KDR | P35968 | Vascular endothelial growth factor receptor 2 | 0.91 | 28 |
| 14 | KIT | P10721 | Stem cell growth factor receptor | 0.53 | 28 |
| 15 | PGR | P06401 | Progesterone receptor | 0.22 | 28 |
| 16 | IL2 | P60568 | Interleukin-2 | 0.42 | 28 |
| 17 | MCL1 | Q07820 | Induced myeloid leukemia cell differentiation protein Mcl-1 | 0.44 | 28 |
| 18 | PIK3R1 | P27986 | PI3-kinase p85-alpha subunit | 0.9 | 28 |
| 19 | MMP2 | P08253 | Matrix metalloproteinase 2 | 0.41 | 27 |
| 20 | IGF1R | P08069 | Insulin-like growth factor I receptor | 0.31 | 26 |
| 21 | CDK4 | P11802 | Cyclin-dependent kinase 4 | 0.12 | 26 |
| 22 | CDK2 | P24941 | Cyclin-dependent kinase 2 | 0.19 | 26 |
| 23 | MET | P08581 | Hepatocyte growth factor receptor | 0.42 | 25 |
| 24 | CASP9 | P55211 | Caspase-9 | 0.61 | 24 |
| 25 | PTPRC | P08575 | Receptor-type tyrosine-protein phosphatase C | 0.29 | 24 |
| 26 | PPARG | P37231 | Peroxisome proliferator-activated receptor gamma | 0.23 | 24 |
| 27 | PARP1 | P09874 | Poly (12) polymerase-1 | 0.42 | 24 |

**3.2.3 Pathway validation of the effects of Baicalin and Wogonin in Hela cells.** The mRNA of EGFR, TNF- α and VEGFA, PI3K and AKT, and MDM 2 gene were detected for 72 h after culture of different experimental and control Hela cells by RT-PCR, and the protein expression was determined by Western blot. The results showed that the mRNA corresponding to the control group, TNF- α, VEGFA, the mRNA (the Fig 11A and B), and protein expression, Fig 11C and D), and with the increase of the drug concentration, the decrease was more obvious. Moreover, under the same drug dose condition, the effect of Wogonin group was more obvious than Baicalin, and the effect of double drug group was more obvious than that of single drug group.

**3.2.4 Potential mechanisms of Baicalin and Wogonin against cervical cancer.** PI3K-AKT signaling pathway is a classical signaling pathway in cells. PI3K is a proto-oncogene whose chemical function can phosphorylate phospholithphthalinositol and its derivatives to generate phospholithphthalinositol 3,4, 5 one triphosphate (PIP3). Epithelial interstitial transformation (EMT) is a biological process in which epithelial cells are transformed into cells with a stromal phenotype through specific procedures. EMT enables tumor cells to acquire the ability to infiltrate and transfer into surrounding tissues. PIP3 Bring domain proteins such as AKT 1 and PDPK 1 to the cell membrane, downregulate the expression of MDM 2 gene and then inhibit the effect of epithelial-mesenchymal transition, thus

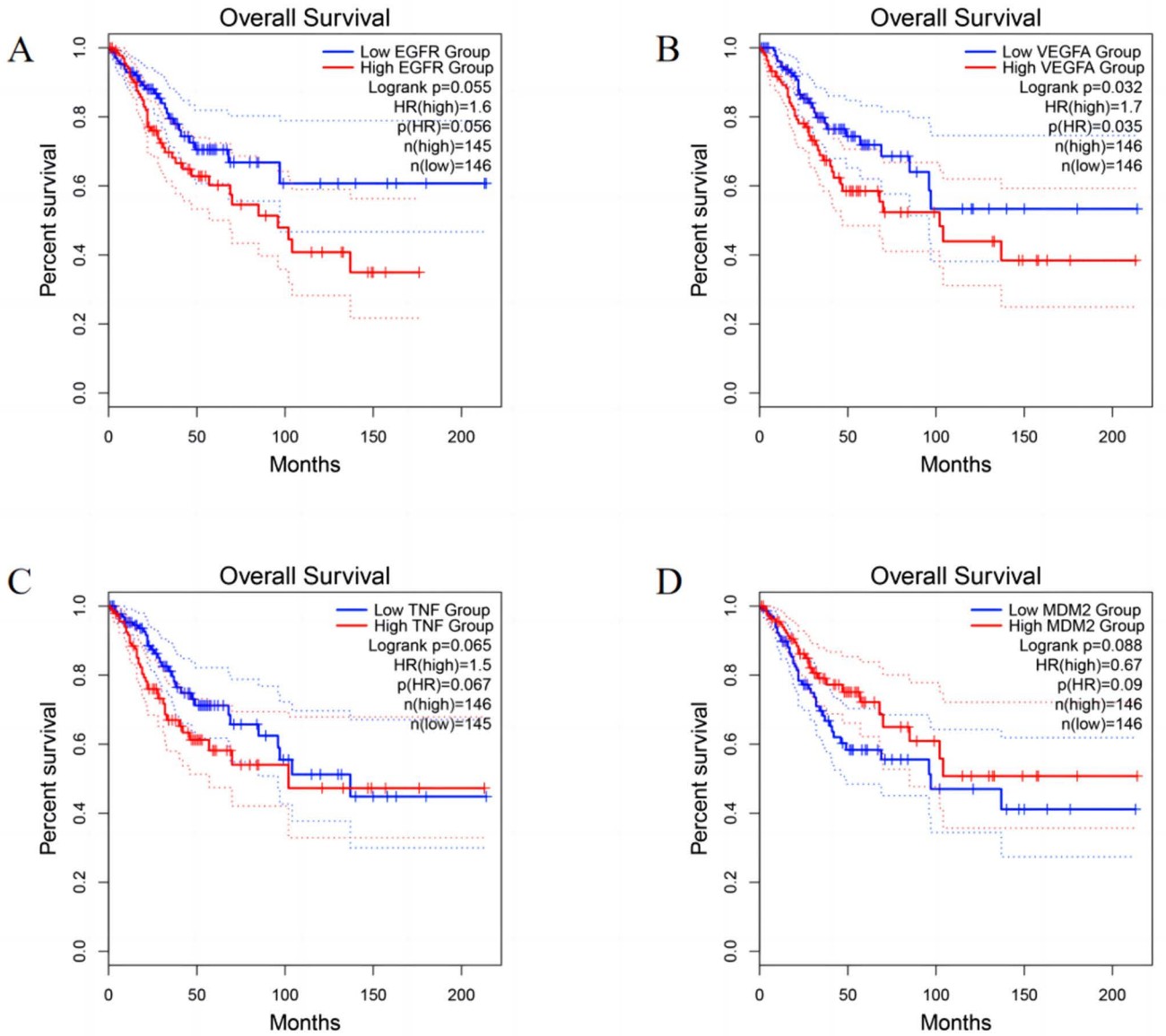

**Fig 6. Core targets for single factor survival analysis:** (A) The relationship between EGFR expression and overall survival of cervical cancer patients; (B) The relationship between VEGFA expression and overall survival of cervical cancer patients; (C) The relationship between TNF expression and overall survival rate of cervical cancer patients; (D) The relationship between MDM2 expression and overall survival in patients with cervical cancer.

activating the signaling cascade of cell growth, proliferation, motility and morphology at the corresponding level to play a key role. As shown in the above results, according to the prediction of network pharmacology, EGFR, TNF- α, VEGFA, PI3K, AKT were selected as the study indicators, confirming that the mechanism of anti-cervical proliferation, migration and apoptosis may be related to the inhibition of PI3K-Akt signaling, and then the inhibition of epithelial-mesenchymal transformation by inhibiting MDM 2 expression. The progression of cervical cancer and chemotherapy resistance are also closely associated with the overactivation of PI3K-AKT pathway, which can exacerbate the proliferation and migration of cervical cancer cells and inhibit the apoptosis of cervical cancer cells. After combining with the EGFR,

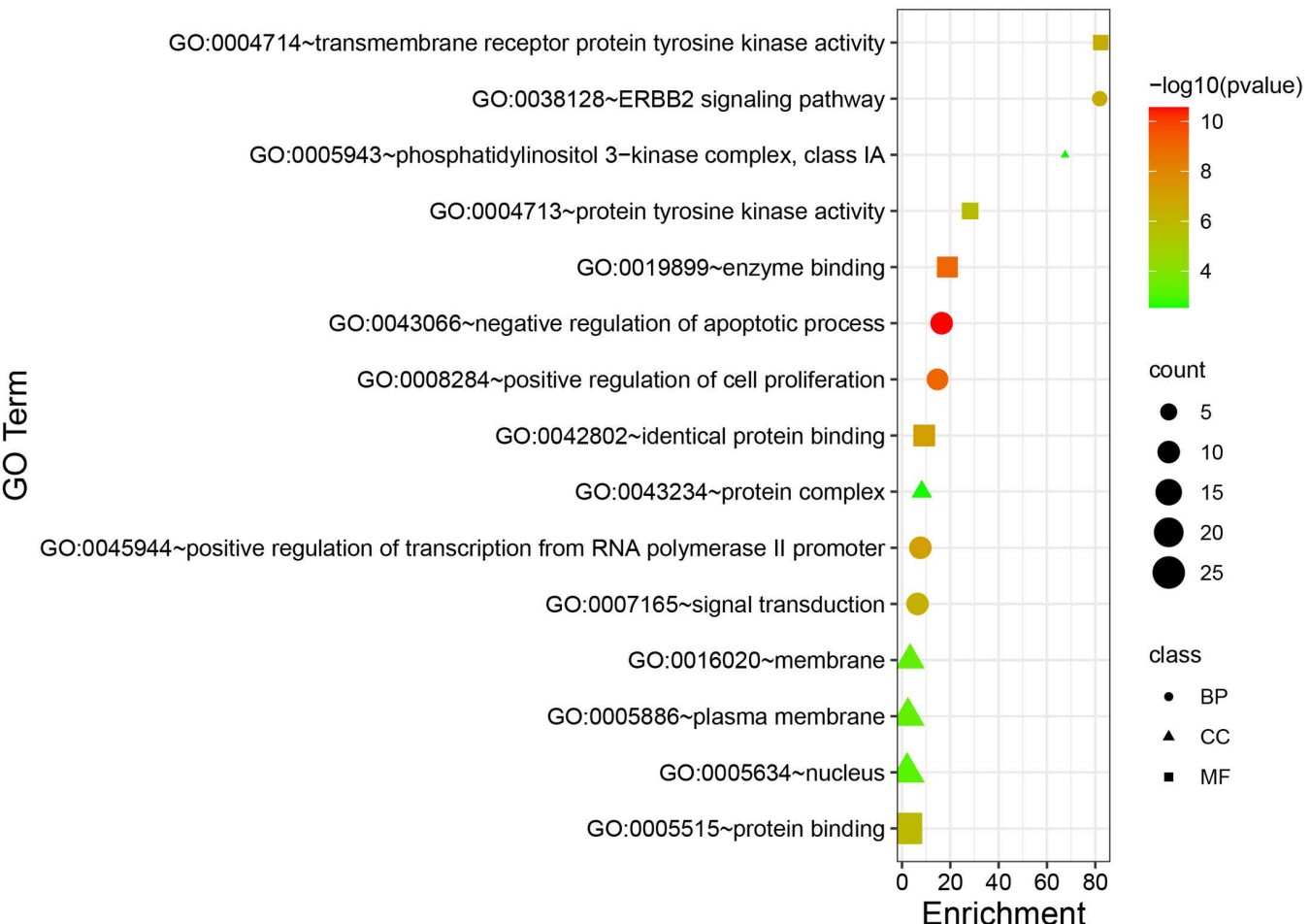

**Fig 7. GO function enrichment analysis.**

TNF-α and VEGFA hub gene targets of cervical cancer, the active ingredients of Scutellaria Barbata were involved in the regulation of PI3K-AKT signaling pathway and the down-regulation of the expression of MDM2 genes and thus inhibited the epithelial-mesenchymal transition, which effectively inhibited the proliferation, migration and promoted the apoptosis of cervical cancer cells (Fig 12).

## 4. Discussion

Chinese herb Scutellaria Barbata has the effect of clearing heat and removing toxicity, removing blood stasis and diuresis. It is often used to treat primary liver cancer, gastrointestinal cancer, nasopharyngeal carcinoma and breast carcinoma ect, and its external use can treat boils and bruises [17,18]. Chemical composition studies showed that the main active ingredients of Scutellaria Barbata are quercetin, lignan, baicalin, wogonin, etc. Polysaccharides, volatile oils, flavonoids, and diterpenoids are the main chemical components [19–21]. Pharmacological studies have shown that Scutellaria Barbata has pharmacological activities such as antitumor, antioxidant, antibacterial and immunomodulatory [22,23]. In this study, using network pharmacology, we found that 14 chemical components may be the active components of Scutellaria Barbata against cervical cancer, with 222 targets related to these 14 major components. Through the cervical cancer disease target mapping, found that 761 targets, intersection

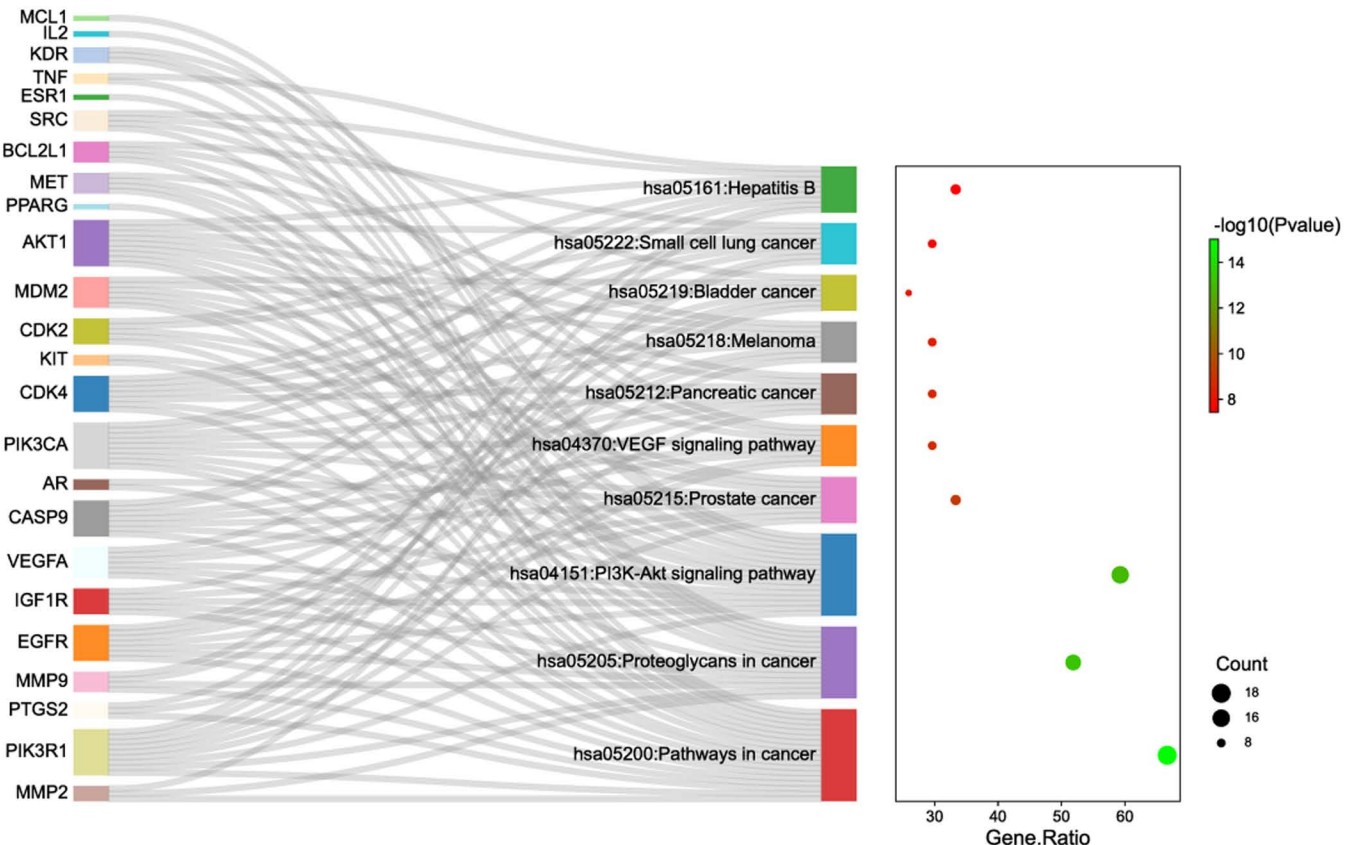

**Fig 8. KEGG functional enrichment analysis.**

of 59, the key target PPI network topological characteristics analysis found 27 core targets, by "Scutellaria Barbata-components-target-cervical cancer" network visible components and target of one-to-many complex relationship, it also just verified the Scutellaria Barbata components, multiple target characteristics. GO and KEGG enrichment analysis of key targets predicted the cancer pathway, PI3K-AKT-MDM2 expression pathway, VEGFA signaling pathway, and TNF signaling pathway for cervical cancer.

Limited to network pharmacology, the stock of data for traditional Chinese medicine research is small, and some data are one-sided, especially the data related to Scutellaria Barbata, which leads to the limitation of prediction results. Therefore, necessary biological experiments are needed to further verify the relevant mechanisms of network pharmacology prediction and mining.

Scutellaria Barbata has shown great potential in antitumor treatments, particularly for cervical cancer in recent years [24]. The antitumor and antiviral effects of Chinese medicine are increasingly attracting the attention of modern medicine, which is important for revealing the mechanism of antitumor effects of Chinese medicine more deeply and screening new antitumor drugs [25]. In order to improve the cure rate and survival rate, in addition to striving to eradicate the disease and improve the survival rate, the treatment should also strive to preserve the reproductive endocrine function of patients when possible, especially for young patients [26]. Flavonoids have good antitumor activity [27]. Quercetin, baicalin, lignan, and apigenin can promote apoptosis of cervical cancer cells and inhibit tumor proliferation and invasion [13]. Various pharmacological activities of Saccharomyces cerevisiae have been

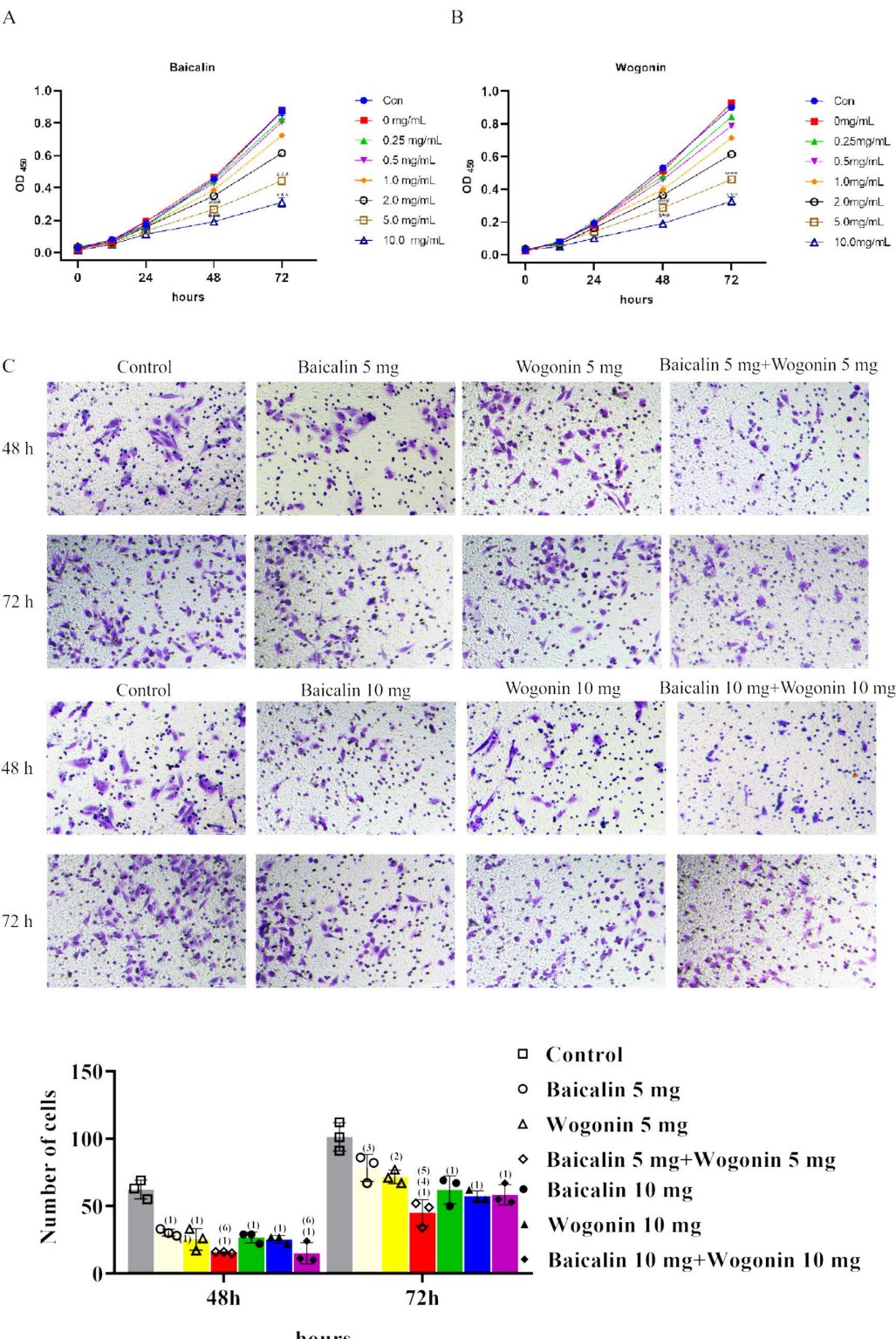

**Fig 9. Detection of proliferation and migration of Hela cells:** (A) The effects of different concentrations of Baicalin on Hela proliferation were detected by CCK8 at different time points; (B) The effect of different concentrations of Wogonin on Hela proliferation detected by CCK8 at different time points; (C) Transwell Migration assay to determine the effect of Baicalin and Wogonin on Hela cell migration at different concentrations.

Table 3. Apoptosis rate of Hela cells in each group (Mean±SD).

| Group | 5mg/ml | 10mg/ml |
|---|---|---|
| Baicalin | 0.20±0.03[(1)] | 0.45±0.07[(1)(3)] |
| Wogonin | 0.38±0.04[(1)] | 0.58±0.03[(1)] |
| Bai-calin + Wogonin Control | 0.46±0.11[(1)] – | 0.65±0.06[(1)(4)] – |

discovered in plants such as antioxidant, anti-inflammatory, antitumor, and neuroprotective: saccharomyces cerevisiae prevents 1,2-dimethylhydrazine-induced experimental colon cancer in male albino Wistar rats; saccharomyces cerevisiae inhibits the transformation of mouse epithelial cells JB6C141 to cancer cells. Gallopanaxanthin inhibited colon cancer cells [28,29]. It has been shown that baicalein can inhibit HeLa cell proliferation and invasion, and its inhibitory effect may be related to down-regulation of COX-2 and VEGF protein expression [30]. In addition, Baicalein exerted a good inhibitory effect on the proliferation and migration invasion of MDA-MB-231 and MCF-7 human breast cancer cells at the in vitro cellular level, and exerted its oncogenic effect through the classical Wnt/β-catenin signaling pathway [31–33]. Studies have shown that the p38/JNK/ERK pathway can inhibit the proliferation, migration and invasion of cervical cancer cells and promote the apoptosis of cervical cancer cells [34,35].

Two common active components of Baicalin and Wogonin were selected from the 14 active components to further verify the results of network pharmacology analysis [13]. Hela cells were selected as the experimental subjects in vitro. The stimulation concentrations of Baicalin and Wogonin were set to 0, 0.25, 0.5, 1, 2, 5 and 10 mg/mL, and the proliferation of Hela cells was detected at 0, 24, 48 and 72 h after stimulation, respectively. We found that Baicalin and Wogonin had an inhibitory effect on the proliferation of Hela cells, and with the increase of drug concentration and stimulation time, the inhibitory effect was more obvious than that of the control group, and the inhibitory effect of the two active ingredients on the growth of Hela cells was concentration - and time-dependent. According to the results of CCK8 test, we selected 5 and 10 mg/mL concentration as the stimulation concentration, and 48 and 72 h as the stimulation time to continue to detect the effects of Baicalin and Wogonin on the migration and apoptosis of Hela cells. The results showed that Baicalin and Wogonin could significantly inhibit the migration of Hela cells, and when the two active ingredients were stimulated at the same time, the inhibitory effect was more obvious than that of single drug stimulation, but there was no statistically significant difference in the inhibitory efficiency of migration between the 5 mg and 10 mg concentration groups. When 5 mg and 10 mg Baicalin and/or Wogonin were used to stimulate Hela cells for 72 h, we found that compared with the control group, the apoptosis rate in the drug stimulation group was significantly increased, and the number of apoptotic cells in the high concentration group was significantly higher than that in the low concentration group, and the apoptosis rate was also higher when the two drugs were simultaneously stimulated than when the single drug was stimulated. Therefore, we believe that Baicalin and Wogonin can effectively inhibit the proliferation and migration of cervical cancer cells, and enhance the apoptosis of cells. Some of the targets with strong docking ability in this study, such as EGFR, MDM2, TNF, and VEGFA, were clearly linked to cervical carcinogenesis and development. Cytochrome P450 is involved in a variety of human metabolic functions, and the genetic polypeptide nature of MDM2 may play a role in the occurrence and development of cervical cancer [36]. TNF and VEGFA are more commonly seen in studies of drugs that promote and inhibit sex hormone secretion, and EGFR inhibits gonadal axis

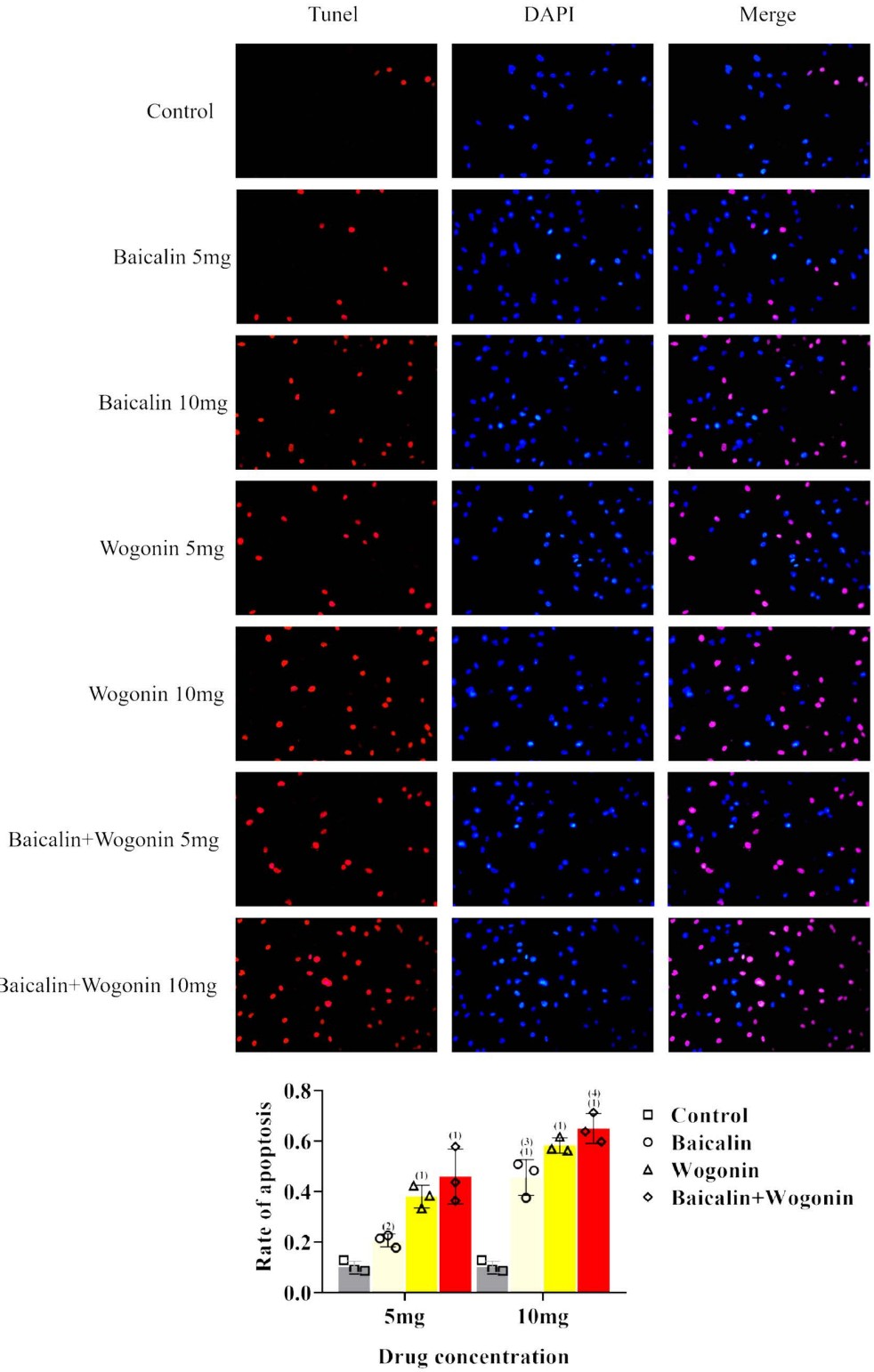

**Fig 10. Detection of apoptosis of Hela cells.** Note:(1) Compared with the control group, *P* < 0.001; (2) Compared with the control group at the same time point, *P* < 0.05; (3) Compared with Baicalin 5 mg group, *P* < 0.05; (4) Compared with Baicalin 10 mg group, *P* < 0.001.

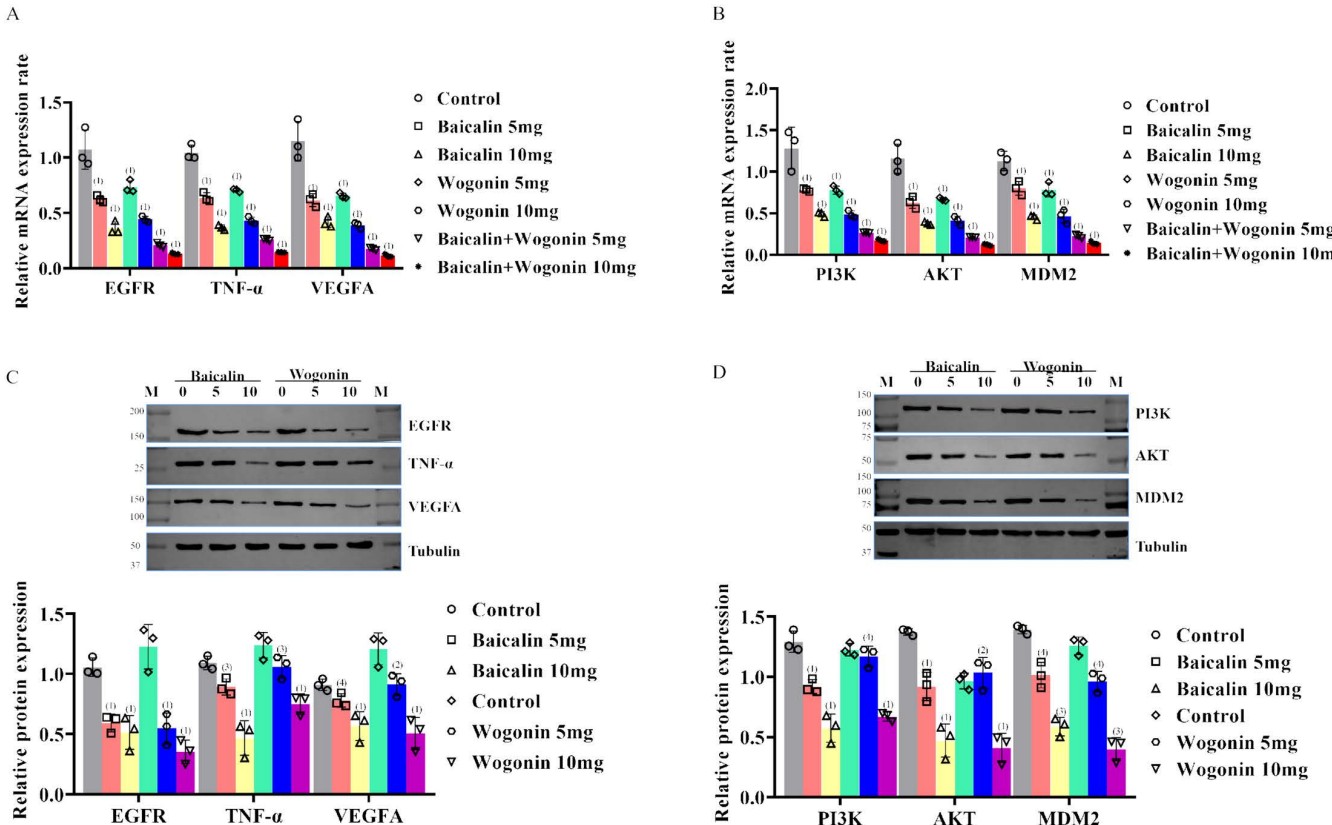

**Fig 11. mRNA and protein expression of target sites and pathways of effective components of Scutellaria Barbata** :(A) mRNA expression of target sites after stimulation of Hela cells with different concentrations of Baicalin and/or Wogonin; (B) mRNA expression of the effect pathway after stimulation of Hela cells with different concentrations of Baicalin and/or Wogonin; (C) Expression of target protein after stimulation of Hela cells with different concentrations of Baicalin and/or Wogonin; (D) Expression of effector pathway protein after stimulation of Hela cells with different concentrations of Baicalin and/or Wogonin. Note: (1) Compared with the corresponding control group, $P < 0.001$; (2) Compared with the corresponding control group, $P < 0.01$; (3) Compared with the corresponding control group, $P < 0.05$; (4) Compared with the corresponding control group, $P > 0.05$.

secretion in breast cancer patients [37]. EGFR levels are not associated with the risk of death and recurrence of cervical cancer, suggesting EGFR is not related to the mechanism of cervical carcinogenesis and progression [38]. Compound activity is closely related to pharmacokinetic properties, suggesting that molecular docking can only be an effective means of screening active ingredients, and subsequent in vivo and ex vivo experiments are still needed for validation [39]. In recent years, there have been remarkable achievements in TCM research on gynecological diseases, but there are still many scientific challenges that need to be solved [40]. In this study, we aimed to investigate the molecular mechanism of action of the active chemical components of Scutellaria Barbata on the VEGF involved pathway in cervical cancer through systematic pharmacological data mining, and to provide a strong basis for subsequent drug development or cervical cancer treatment [41]. Existing findings have been elaborated for cervical cancer VEGF expression mechanisms and drug mechanism of action studies, but not in depth, in a retrospective cohort study of 64 patients, protein expression was obtained by immunohistochemistry from biopsies containing tumors and stroma [42]. It has been demonstrated that miR-125 as affects a variety of cancer-related and to expression, which is downregulated in cervical cancer tissues and cell lines, while VEGF is upregulated [43]. Negatively regulates VEGF expression in cervical cancer tissues by targeting and regulating VEGF to inhibit proliferation, invasion and migration in cervical cancer [44]. It was demonstrated

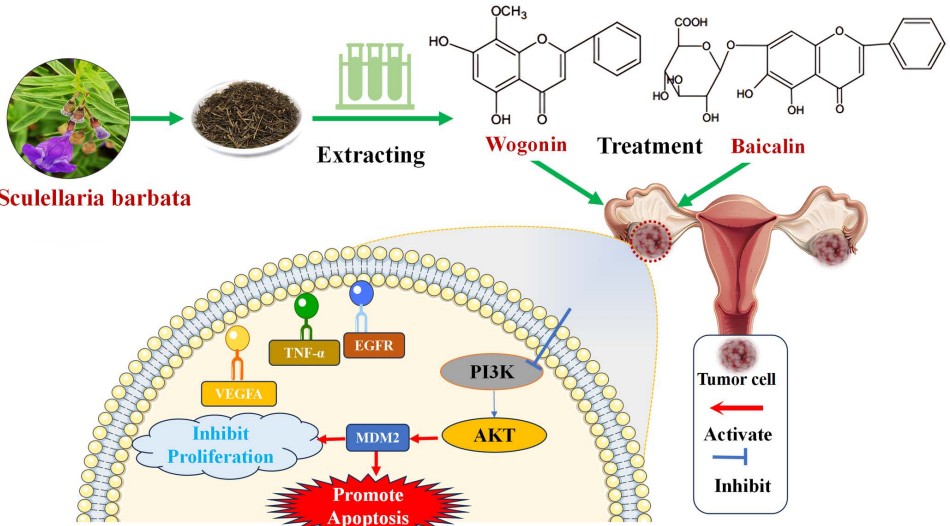

**Fig 12. Targets and potential mechanisms of pathways for the action of effective components of Sculellaria Barbata annuus in cervical cancer.**

that miR-520d-5p inhibited the activation of PI3K-AKT signal pathway, thereby suppressing the progression and development of cervical cancer [43]. Numerous studies have shown that, EGFR is an oncogene in cervical cancer, The rise of EGFR is closely related to the occurrence and development of cervical cancer, While our findings suggest that, Most likely, by inhibiting EGFR expression, Thus to play a role in the treatment of cervical cancer, And suggested in the KEGG pathway enrichment analysis, The PI3K-AKT-related pathway may be an important pathway affecting the development of cervical cancer, In combination with the preexisting studies, It cannot be excluded that by inhibiting the expression of EGFR, Affect the PI3K-AKT related pathway and thus decrease the proliferation and migration of cervical cancer cells, Thus to play a therapeutic role in cervical cancer.

Tumor necrosis factor (TNF),as a member of the TNF superfamily of cytokines, mediates the processes of cellular inflammation, differentiation, proliferation and apoptosis [45]. And the current reference to TNF generally refers to TNF-α accounting for about 70%-95% of the total TNF bioactivity [46].TNF plays different regulatory roles in different diseases such as coronary heart disease, rheumatoid arthritis, cancer, and ankylosing spondylitis, and especially its anti-tumor effects have received much attention in recent years. It has been found that the malignancy of cervical cancer is positively correlated with cellular TNF expression, and therefore, cellular TNF-α expression can be used as an observational index for cervical cancer grading as well as prognosis [47]. TNF-α plays an important role in the induction of cervical cancer cell proliferation, and it has also been demonstrated that TNF-α is correlated with cervical cancer in the elderly.The TNF signal pathway is an important inflammatory response signal pathway, in which the associated factor receptors can also induce cellular regulation [48]. The MAPK signal pathway is one of the classical signal pathways, and studies have shown that this pathway promotes cervical cancer angiogenesis by regulating the expression of VEGF vascular-related factors, and conversely, blocking this pathway inhibits the proliferation and angiogenesis of cervical cancer cells [49]. In our in vitro validation experiment, we verified the potential targets, pathways and target genes of the above active components. The results showed that both of the two active components stimulated little and could decrease the mRNA and protein expressions of the three possible targets of EGFR, TNF-α and VEGFA in Hela cells. The expression of two

signal pathway proteins and their downstream target genes, which were speculated by network pharmacology, were also decreased, suggesting that the active components of Scutellaria Barbata were likely to regulate the PI3K-AKT signaling pathway by acting on the three targets of EGFR, TNF-α and VEGFA, and down-regulating the expression of MDM2 gene. Therefore, the proliferation and migration of Hela cells were inhibited and the apoptosis of Hela cells was enhanced.

## 5. Conclusion

In conclusion, Scutellaria Barbata treats cervical cancer mainly through multiple components, multiple targets, and multiple pathways. EGFR, TNF-α, and VEGFA are the key targets related to the prognosis of cervical cancer. The effective components of Scutellaria Barbata can regulate the PI3K-AKT signaling pathway and down-regulate the expression of MDM2 gene to inhibit epithelial-mesenchymal transition, thus effectively inhibiting proliferation, migration, and promoting apoptosis. This provides a reference for further research.

## Supporting information

**S1 Table. Sequences of primers used for real-time RT-PCR.**
(PDF)

## Acknowledgments

We are grateful for TCMSP/STITCH/DisGeNET websites for providing access to the data. Thanks to GeneCard database and DrugBank database for free use.

## Author contributions

**Conceptualization:** Wenqi Feng, Gang Tian.

**Data curation:** Wenqi Feng, Guobing Wang, Dexin Wang, Congchao Jia, Lan Li, Gang Tian.

**Formal analysis:** Guobing Wang, Congchao Jia, Lei Liang, Lan Li, Gang Tian.

**Funding acquisition:** Guobing Wang, Gang Tian.

**Investigation:** Guobing Wang, Dexin Wang, Congchao Jia, Lan Li, Gang Tian.

**Methodology:** Guobing Wang, Dexin Wang, Congchao Jia.

**Project administration:** Wenqi Feng, Guobing Wang, Gang Tian.

**Resources:** Wenqi Feng.

**Software:** Guobing Wang, Dexin Wang, Congchao Jia, Lei Liang, Lan Li.

**Supervision:** Wenqi Feng, Guobing Wang, Lan Li, Gang Tian.

**Validation:** Guobing Wang, Dexin Wang, Congchao Jia, Lei Liang, Lan Li.

**Visualization:** Wenqi Feng, Dexin Wang, Lei Liang, Gang Tian.

**Writing – original draft:** Guobing Wang, Dexin Wang.

**Writing – review & editing:** Wenqi Feng, Congchao Jia, Gang Tian.

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
