## [Decision Letter · Decision Letter 0]

22 Nov 2024

PONE-D-24-33635Scutellaria Barbata inhibits epithelial-mesenchymal transformation through PI3K/AKT and MDM2 thus inhibiting the proliferation, migration and promoting apoptosis of Cervical Cancer cellsPLOS ONE

Dear Dr. feng,

Thank you for submitting your manuscript to PLOS ONE. After careful consideration, we feel that it has merit but does not fully meet PLOS ONE’s publication criteria as it currently stands. Therefore, we invite you to submit a revised version of the manuscript that addresses the points raised during the review process.

We look forward to receiving your revised manuscript.

Kind regards,

Shahbaz Ahmad Zakki

Academic Editor

PLOS ONE

Journal Requirements:

 “This study was supported by grants from the Luzhou Science and Technology Department Applied Basic Research Program (No: 2022-JYJ-145), the Sichuan Province Science and Technology Department of foreign (border) high-end talent introduction project (No: 2023JDGD0037), Sichuan Provincial Medical Association (No: Q22027), YiBin Science and Technology Department Social Development Projects (No:2022SF004), and Medical Scientific Research Project of Yibin Municipal Health Commission (No:2023YW026).”        

Reviewers' comments:

Reviewer's Responses to Questions

**Comments to the Author**

1. Is the manuscript technically sound, and do the data support the conclusions?

Reviewer #1: Yes

Reviewer #2: Yes

2. Has the statistical analysis been performed appropriately and rigorously? 

Reviewer #1: Yes

Reviewer #2: Yes

3. Have the authors made all data underlying the findings in their manuscript fully available?

Reviewer #1: Yes

Reviewer #2: Yes

4. Is the manuscript presented in an intelligible fashion and written in standard English?

Reviewer #1: Yes

Reviewer #2: No

5. Review Comments to the Author

Reviewer #1: This manuscript covered all the aspect and is of good quality. there are some grammatical mistakes and is recommended for publication. I thoroughly go through this manuscript, and it will fulfill all the requirements for publications

Reviewer #2: In the manuscript entitled “Scutellaria Barbata inhibits epithelial-mesenchymal transformation through

PI3K/AKT and MDM2 thus inhibiting the proliferation, migration and promoting

apoptosis of Cervical Cancer cells”, wen-qi feng et al have presented effect of Baicalein and Wogonin which are active compounds of Scutellaria Barbata. Overall, this work builds nicely on the previous reports about

the anti-tumor effects of Scutellaria Barbata, though there are some experimental issues that should be addressed, and some editing is required to better contextualize the meaning of these results. Discussion section should be improved. Authors are commended on assembling interesting scientific data; I look forward to reviewing a revised version of the manuscript soon.

Comments:

(1) Lines#33-37: Authors should re-write the sentences.

(2) If possible, date of access of websites for screening of active component should be mentioned.

(3) Method section 2.1.2-2.1.3 (Lines# 111-121): The English is very poor. It is better to improve whole sections.

(4) Line 134: Authors should correct degree(degree).

(5) Method section 2.2: It is better to add method that how authors prepared and isolated component from the plant.

(6) Line#155: Please check and correct the sentence if required.

(7) Lines#165-166: Did the authors used the medium in liters?

(8) Line#204: It is difficult to understand. Author should change the heading so that readers can understand easily.

(9) Lines#212-213: It is difficult to understand. Authors should make a clear statement regarding common targets.

(10) Lines#237-238: Authors should re-write the outcomes of the experiments.

(11) Line#242: (after the addition of baicalin and baicalein,). Authors should correct it. Outcomes should be re-written.

(12) Line#246: There is confusion of active components. Authors should clarify baicalin and bacalealin. It is mentioned multiple times in the manuscript.

(13) Table 3: Table 3 should be divided into 3 parts. So that readers can understand easily. Or authors should prepare it again in a way that it is easy to understand.

(14) Line#284: This line should be checked and corrected.

(15) Conclusion section also needs to be improved.

(16) English language was ok but could be improved in some places. Large language models, such as chatGPT could be useful for the fluidity of the language, assuming that the authors meticulously check for the veracity of the statements.

6. PLOS authors have the option to publish the peer review history of their article (what does this mean? ). If published, this will include your full peer review and any attached files.

**Do you want your identity to be public for this peer review?** For information about this choice, including consent withdrawal, please see our Privacy Policy .

Reviewer #1: **Yes: ** Dr Abdul Jabbar

Reviewer #2: No

---

## [Author Response · Author response to Decision Letter 1]

22 Jan 2025

1.Lines#33-37: Authors should re-write the sentences.

1.Reply: Thank you very much for you valuable comments and suggestion. According to your suggestion, we rewrote these sentences as follows: Tunel results showed that single-agent Baicalein or Wogonin with 10 mg/mL had the highest rate of apoptosis on Hela cells after culturing the cells for 72 h. Baicalein or Wogonin (10 mg/mL + 10 mg/mL) had the highest apoptosis ratio in Hela cells after 72h and a higher combination ratio than in the single agent group (P <0.001).

2.If possible, date of access of websites for screening of active component should be mentioned.

2.Reply: Thank you for your suggestions. The majority of articles do not mention database access time in the main body, and we can provide the raw data if needed.

3.Method section 2.1.2-2.1.3 (Lines# 111-121): The English is very poor. It is better to improve whole sections.

3.Reply: Thank you very much for you valuable comments and suggestion. According to your suggestion, we have rewritten this part：

2.1.2 Acquisition of targets of action in cervical cancer

The human Gene-disease-related database, OTP (https://www.opentargetsplatform.org/), and GeneCards (https://www.genecards.org/) offer comprehensive information on all known and predicted human genes in various aspects such as the genome, proteome, transcription, genetics, and function. In this study, we conducted a search using the GeneCards database and the OTP database to identify targets associated with cervical cancer disease. The results from both databases were merged, structured, and refined based on a predefined threshold to pinpoint the targets relevant to cervical cancer.

2.1.3 Construction of a drug-active ingredient-target gene-disease network

The target genes corresponding to the active ingredients of Scutellaria Barbata and the target genes related to cervical cancer were introduced into venny2.1.0 (https://bioinfogp.cnb.csic.es/tools/venny/) to obtain the intersection between the two, and the common genes of the two were the key targets of Scutellaria Barbata against cervical cancer. The Cytoscape3.7.2 software was used to construct the "drug-active ingredient-target gene-disease" relationship network to explore the mechanism of Scutellaria Barbata action against cervical cancer.

4.Line 134: Authors should correct degree(degree).

4.Reply: Thank you for kindly reminding us，and we are very sorry for this error. We correct the degree as follow：Topological analysis of the network was performed using the Network Analyzer plugin to screen out the top 10 core targets for Scutellaria Barbata treatment cervical cancer based on the degree (degree ≥31) values.

5.Method section 2.2: It is better to add method that how authors prepared and isolated component from the plant.

5.Reply: Thank you for your feedback. Both baicalin and wogonin were purchased from Sigma-Aldrich (USA) and have been duly acknowledged in the manuscript (Line#156).

6.Line#155: Please check and correct the sentence if required.

6.Reply: Thank you for your feedback. We have revised the sentence as follows:

Cells were cultured at 37℃, 5% CO2, and 95% air in a medium comprising 90% DMEM medium (Thermo Fisher, USA), 10% fetal bovine serum (Thermo Fisher, USA), and 1% penicillin-streptomycin.

7.Lines#165-166: Did the authors used the medium in liters?

7.Reply: Thank you for your feedback. Upon reviewing the article, we identified an error in the unit notation and have corrected it as follows:

In the upper chamber, 1×105 cells were seeded in 200 μL of serum-free medium, while 800 μL of complete medium, serving as the chemo attractant, was added simultaneously to the lower chamber.

8.Line#204: It is difficult to understand. Author should change the heading so that readers can understand easily.

8.Reply: We agree with the comment and rewrote the heading in the revised manuscript as the following: Construction of “ Scutellaria Barbata - active ingredient - target gene - cervical cancer ” network relationship

9.Lines#212-213: It is difficult to understand. Authors should make a clear statement regarding common targets.

9.Reply: Thank you for pointing out this probelm in manuscript, we have rewritten this part and picture of core targets have been supplemented as follows:

String database is a protein interaction network database based on public database and literature information, which can provide comprehensive information of protein interaction.The common target genes of Scutellaria Barbata and uterine neck cancer were entered into String database for PPI analysis, and then the obtained data were imported into Cytospace, and 27 core targets were obtained according to degree ≥ 24.(Figure 5 and Table 2).

10. Lines#237-238: Authors should re-write the outcomes of the experiments.

10.Reply: Thank you for your feedback. We have revised the experimental results section based on your suggestions：

The study confirmed that the active ingredients baicalin and wogonin from Scutellaria barbata can effectively inhibit the proliferation of Hela cells. CCK8 assay results showed a significant decrease in cell proliferation in the experimental group compared to the control group at 48 h and 72 h (P<0.001). This effect was most pronounced at a drug concentration of 10 mg/mL (P<0.001). Over a 72-hour period, there was a significant increase in the inhibition of Hela cell proliferation with both the passage of time and escalating drug concentrations (P<0.01). The suppression of Hela cell proliferation was more pronounced when the drug concentrations reached 5 mg/mL and 10 mg/mL (Figure 9A, B). In the Transwell migration assay, the experimental group exhibited significantly reduced cell migration compared to the control group. Furthermore, simultaneous stimulation with baicalin and wogonin at the same time point resulted in a more pronounced inhibition of cell migration compared to single-drug treatment (P<0.001, Figure 9C).

11.Line#242: (after the addition of baicalin and wogonin,). Authors should correct it. Outcomes should be re-written.

11.Reply: We appreciate your input, and have made revisions to the section as suggested.

12.Line#246: There is confusion of active components. Authors should clarify baicalin and bacalealin. It is mentioned multiple times in the manuscript.

12.Reply: Thank you for bringing this to our attention. "Baicalealin" is not the standard term; its presence in the manuscript may be due to a spelling error. We have replaced all instances of "Baicalealin" with "Baicalin.

13. Table 3: Table 3 should be divided into 3 parts. So that readers can understand easily. Or authors should prepare it again in a way that it is easy to understand.

13.Reply: Thank you for your suggestions. We have redrawn Table 3, and the results are as follows:

Table 3 Apoptosis rate of Hela cells in each group（Mean±SD）

Group 5mg/ml 10mg/ml

Baicalin 0.20±0.03(1) 0.45±0.07(1)(3)

Wogonin 0.38±0.04(1) 0.58±0.03(1)

Baicalin+ Wogonin

Control 0.46±0.11(1)

- 0.65±0.06(1)(4)

-

14.Line#284: This line should be checked and corrected.

14.Reply: We have re-written this part according to the reviewer’s comments.

Chinese herb Scutellaria Barbata has the effect of clearing heat and removing toxicity, removing blood stasis and diuresis. It is often used to treat primary liver cancer, gastrointestinal cancer, nasopharyngeal carcinoma and breast carcinoma ect, and its external use can treat boils and bruises(17, 18).

15.Conclusion section also needs to be improved

15.Reply: we tried our best to improve this section, and this changes will not influence the content and framework of the paper.

In conclusion, Scutellaria Barbata treats cervical cancer mainly through multiple components, multiple targets, and multiple pathways. EGFR, TNF-α, and VEGFA are the key targets related to the prognosis of cervical cancer. The effective components of Scutellaria Barbata can regulate the PI3K-AKT signaling pathway and down-regulate the expression of MDM2 gene to inhibit epithelial-mesenchymal transition, thus effectively inhibiting proliferation, migration, and promoting apoptosis. This provides a reference for further research.

16. English language was ok but could be improved in some places. Large language models, such as chatGPT could be useful for the fluidity of the language, assuming that the authors meticulously check for the veracity of the statements.

16.Reply: We appreciate your valuable and sincere feedback, and have utilized AI to enhance certain sentences.

---

## [Editor Report · Decision Letter 1]

9 Mar 2025

Scutellaria Barbata inhibits epithelial-mesenchymal transformation through PI3K/AKT and MDM2 thus inhibiting the proliferation, migration and promoting apoptosis of Cervical Cancer cells

PONE-D-24-33635R1

Dear Dr. Feng

We’re pleased to inform you that your manuscript has been judged scientifically suitable for publication and will be formally accepted for publication once it meets all outstanding technical requirements.

Kind regards,

Shahbaz Ahmad Zakki

Academic Editor

PLOS ONE
---

## [Editor Report · Acceptance letter]

PONE-D-24-33635R1

PLOS ONE

Dear Dr. Feng,

I'm pleased to inform you that your manuscript has been deemed suitable for publication in PLOS ONE. Congratulations! Your manuscript is now being handed over to our production team.

Kind regards,

on behalf of

Dr. Shahbaz Ahmad Zakki

Academic Editor

PLOS ONE